# Microbial Dynamics and Pathogen Control During Fermentation of Distiller Grains: Effects of Fermentation Time on Feed Safety

**DOI:** 10.3390/ijms252111463

**Published:** 2024-10-25

**Authors:** Mingming Zhu, Duhan Xu, Chaosheng Liao, Tiantian Zhang, Bijun Zhou, Kaigong Wang, Ping Li, Zhentao Cheng, Chao Chen

**Affiliations:** College of Animal Science, Guizhou University, Guiyang 550025, China; 18608518394@163.com (M.Z.); l8718227x1@163.com (D.X.); liaocs@yeah.net (C.L.); ztt18286158817@163.com (T.Z.); bjzhou@gzu.edu.cn (B.Z.); as.kgwang@gzu.edu.cn (K.W.); lip@gzu.edu.cn (P.L.); ztcheng@gzu.edu.cn (Z.C.)

**Keywords:** white distiller grains, fermentation duration, microbial communities, pathogenic risk, LC–MS metabolomics, animal feed

## Abstract

Determining the effects of fermentation duration on the microbial ecosystem, potential pathogenic risks, and metabolite generation during the fermentation of distilled grains is essential for safeguarding the safety and enhancing the nutritional profile of animal feed. This study investigates the effect of varying fermentation times (9, 30, and 60 days) on microbial diversity, pathogenic risk, and metabolite profiles in distiller grains using 16S rDNA sequencing and LC–MS-based metabolomics. The results showed that early fermentation (9–30 days) enhanced the abundance of beneficial bacteria, such as *Lactobacillus reuteri* and *Lactobacillus pontis* (*p* < 0.05), while pathogenic bacteria, like *Serratia marcescens* and *Citrobacter freundii*, were significantly reduced (*p* < 0.05). Metabolomic analysis revealed an increase in unsaturated fatty acids and the degradation of biogenic amines during early fermentation. However, prolonged fermentation (60 days) led to a resurgence of pathogenic bacteria and reduced the synthesis of essential metabolites. These findings suggest that fermentation duration must be optimized to balance microbial safety and nutrient quality, with 30 days being the optimal period to reduce pathogenic risks and enhance feed quality.

## 1. Introduction

With the global increase in meat consumption and livestock and poultry farming, feed resources are becoming increasingly scarce. In this context, distiller grain, a byproduct of Chinese liquor brewing, has attracted attention. Distiller grain mainly refers to the residues left after fermenting sorghum, wheat, and other raw materials to extract alcohol and aromatic substances. The main nutrients in distiller grain include proteins, fats, and cellulose, which are all rich in fermentation products, such as active factors [1], making it an excellent candidate for alternative feed. Livestock and poultry find it difficult to digest and absorb fresh distiller grain, resulting in low nutrient utilization. Owing to its high acidity and moisture content, it is highly susceptible to mold. If discarded directly, it not only severely pollutes the environment but also wastes resources [2].

Recent studies have confirmed that microorganisms can convert starch, sugars, cellulose, and other substances in distiller grain into metabolic products, such as oligosaccharides, short-chain fatty acids, and vitamins. They also produce lipase, protease, and cellulase, which improve the digestibility of distiller grain [3]. Many fermentation strains, such as lactic acid bacteria (LAB) and yeasts, are permitted for use in China. These strains can also lower the pH of fermented feed and increase its lactic acid content, which have positive effects on animal gut health and production performance [4]. Some researchers have added strains to distiller grain for fermentation before feeding to overcome the problem of poor digestibility in livestock and poultry. Fermented distiller grains have been proven useful in cattle farming, where consuming feed containing fermented distiller grain promotes the average daily gain and immune capacity of cattle [5]. However, He et al. [5] found that cattle that consume a high proportion of fermented distiller grains have an increased risk of diseases. This suggests that the fermentation of distiller grain may be accompanied by a proliferation of certain pathogenic bacteria (e.g., *Salmonella enterica*), which increases the risk of microbial contamination and pathogen invasion. Few studies have examined the bacterial communities in distiller grains during the dynamic fermentation process, but it is crucial because pathogenic microorganisms during fermentation may be key factors in animal diseases.

Microbial communities and their biochemical metabolic pathways play an irreplaceable role [6]. A full understanding of the metabolic pathways of bacterial communities in the fermentation environment is crucial for improving the quality and safety of fermented distiller grains as feed. However, few studies have examined the metabolism of bacterial communities in distiller grain fermentation systems. In recent years, molecular tools, such as metagenomics, have greatly facilitated the exploration of microbial communities and microbial metabolites in the gastrointestinal tract. A feasible approach is to use metagenomics with liquid chromatography–mass spectrometry (LC–MS) to study the metabolic pathways of microbial communities in the fermentation environment.

This study employs 16S rDNA sequencing and LC–MS technology to investigate trends in microbial community dynamics and metabolite profiles during the fermentation of distiller grains. This study aims to provide theoretical references for a clean and safe production of feed-grade distiller grains and their subsequent application in animal feed studies. The objectives of this study are as follows: (1) to elucidate the patterns of pathogenic bacteria attachment to distiller grains during different fermentation stages; and (2) to determine whether fermentation can effectively reduce the risk of microbial infections in distiller grains.

## 2. Results and Discussion

### 2.1. Characteristics of Bacterial Community Changes During Distiller Grain Fermentation

The changes in bacterial alpha diversity during the fermentation of the distiller grains are shown in Table 1. Good coverage was used to assess sequencing reliability. In this study, coverage for all treatments was above 0.99, demonstrating the reliability of the 16S rDNA sequencing in this research. The Chao1 index and observed species were used to evaluate the abundance of bacterial communities. After fermentation, both the Chao1 index and observed species showed a decreasing trend initially but experienced a sudden increase at day 60 of fermentation. The Shannon and Simpson indices were used to assess the diversity of the bacterial communities. The diversity of bacterial communities showed a decreasing trend in the early stages (9 days) after fermentation but gradually increased in the later stages. For fermented products, the decrease in both bacterial community abundance and diversity after fermentation was attributed to lactic acid bacteria lowering the pH by suppressing the growth of other bacteria [7]. The increase in bacterial community abundance and diversity during the later stages of fermentation may be due to certain bacteria possessing high resistance.

The differences in bacterial composition were assessed using non-metric multidimensional scaling (NMDS) analysis, with a Stress < 0.2 indicating a good model fit (Figure 1). In this study, there were significant differences in the bacterial composition before and after fermentation. At different stages of fermentation, the bacterial composition varied and showed some correlations. As fermentation progressed, the bacterial community gradually became more similar to that before fermentation. Typically, fermentation lowers the pH through lactate production to inhibit the proliferation of undesirable microorganisms, including pathogens. This is a significant factor and contributes to the differences in bacterial composition before and after fermentation. However, in the later stages of fermentation, the microbial community gradually approached pre-fermentation states, which contributed to the proliferation of pathogens in the fermented distiller grains.

### 2.2. Characteristics of Changes in Bacterial Community Composition During Distiller Grain Fermentation

To further understand the changes in pathogenic bacteria during the fermentation of distiller grains, we used RDP classifier v2.2 software from the SILVA database to annotate and summarize the OTUs of each sample at phylum, genus, and species classification levels (Figure 2a). At the phylum level, *Firmicutes* were predominantly observed before fermentation (SJZ), consistent with the findings reported by Zuo et al. [8]. As fermentation progressed, *Firmicutes* gradually increased in abundance and peaked on day 30 of fermentation (SJZ30). Most *Firmicutes* bacteria can produce spores, which allows them to survive in extreme environments, such as high temperatures and water scarcity, and possess strong degradation capabilities [9]. These characteristics are essential for *Firmicutes* to become the dominant phylum in fermented distiller grains.

At the genus level, before fermentation, the predominant genera were *Bacteroides* (7.84%) and *Lactobacillus* (5.10%). As fermentation progressed, *Lactobacillus* gradually increased to become the predominant genus, peaking at day 30, whereas *Bacteroides* gradually decreased during fermentation but suddenly increased at day 60. In the brewing process, the internal microbial community of Daqu in the later stages of fermentation was mainly dominated by *Lactobacillus* [10]. A significant increase in *Lactobacillus* abundance was also a major reason for the increase in the abundance of *Firmicutes*. *Lactobacillus* inhibits the growth and reproduction of other bacteria by secreting substances such as bacteriocins, fatty acids, and polysaccharide–protein complexes [11,12], thereby reducing the levels of pathogenic and spoilage bacteria. This characteristic of *Lactobacillus* leads to a gradual decrease in the abundance of *Bacteroides* during fermentation. However, the abundance of *Bacteroides* increased during the later stages of fermentation. *Bacteroides* possess potent virulence factors and strong resistance, making them one of the main pathogens causing bacteremia and abscesses in body parts [13]. This increase in *Bacteroides* in the later stages of fermentation exacerbates pathogenic risk in fermented distiller grains.

To further understand the changes in pathogenic bacteria during fermentation, bacterial communities were annotated at the species level, and the relative abundances of representative species were compared using one-way ANOVA (Figure 2b). The analysis showed that fermentation increased the relative abundance of beneficial bacteria (*Lactobacillus reuteri, Lactobacillus pontis,* and *Pediococcus acidilactici*), but this increase was limited to the early to middle stages of fermentation (9–30 d) and gradually decreased as fermentation progressed. Especially notable were *Lactobacillus reuteri* and *Lactobacillus pontis,* which promote gut health and enhance immunity in animals [14]. Similarly, fermentation inhibited the increase in the relative abundance of certain pathogenic bacteria, such as *Serratia marcescens* and *Citrobacter freundii* [15]; however, the abundance of these bacteria increased during the later stages of fermentation (60 d). Furthermore, the *Bacteroides vulgatus PC510, Bacteroides ovatus V975,* and *Clostridiales bacterium CIEAF 020* identified in this study showed an increasing trend during the later stages of fermentation. These strains have been associated with inflammation and various diseases [16,17], but more evidence is required to establish the risk of infection in humans and animals. Nevertheless, fermentation promoted an increase in beneficial bacteria and inhibited an increase in pathogenic bacteria. It should be noted that the significant increase in pathogenic bacteria in the later stages of fermentation may increase the risk of livestock infection, but further research is required to confirm this.

### 2.3. Prediction of Pathogenic Risk During Distiller Grain Fermentation

To further understand the pathogenic risk during the distiller grain fermentation process, Ward et al. [18] predicted the bacterial community phenotypes during fermentation (Figure 3). Regarding bacterial characteristics (Figure 3a), fermentation reduced aerobic, anaerobic, and Gram-negative bacteria while increasing facultative anaerobic and Gram-positive bacteria and those containing mobile genetic elements. Fermentation occurs in an anaerobic environment, which gradually diminishes the metabolism of aerobic bacteria [19] and is replaced by anaerobic and facultatively anaerobic bacteria capable of surviving under anaerobic conditions. This explains the decrease in aerobic Gram-negative bacteria (most aerobic bacteria are Gram-negative) and the increase in facultative anaerobic bacteria observed in this study after fermentation.

The low pH generated during fermentation inhibits the proliferation of most anaerobic bacteria, allowing only those capable of tolerating low pH (such as lactic acid bacteria) to thrive. The pH tolerance of lactic acid bacteria contributes to their high relative abundance after fermentation, which also explains the decrease in anaerobic bacteria and increase in Gram-positive bacteria. “Contains mobile elements” refers to mobile genetic elements within bacterial genomes, including transposons, plasmids, and phages, which play crucial roles in bacterial adaptation to new environments and in the acquisition of new metabolic capabilities [20]. Their increase is a result of bacterial adaptation to the low pH environment during distiller grain fermentation.

Regarding bacterial pathogenicity (Figure 3b), fermentation reduced the abundance of potentially pathogenic bacteria, biofilm-forming bacteria, and stress-tolerant bacteria. Traditionally, fermentation inhibits bacterial proliferation, resulting in a simplified bacterial network in the fermentation system [21]. Simple networks lack stability and adaptability to the external environment of complex networks. Consequently, fermentation in this study resulted in a decrease in the number of stress-tolerant bacteria. Similarly, fermentation suppressed the growth of potentially pathogenic bacteria. Additionally, a decrease in the relative abundance of Gram-negative bacteria and biofilms, accompanied by a reduction in the relative abundance of potentially pathogenic bacteria, is advantageous for the fermented products. Gram-negative bacteria with biofilms, such as lipopolysaccharide layers, are challenging and are reported to cause severe systemic infections [22]. In this study, Gram-negative bacteria and bacteria that form biofilms showed higher relative abundances at 60 days of fermentation compared to 30 days; potentially pathogenic bacteria exhibited a similar trend. This change may be associated with the increase in pathogenic bacteria. Despite fermentation reducing potential pathogenic risks, the fact that these risks increased in the later stages of fermentation cannot be ignored. Therefore, we recommend using fermented distiller grains for animal feed within 30 days of fermentation to significantly reduce the risk of animal disease.

### 2.4. Changes in Metabolites During the Distiller Grain Fermentation Process

To ensure the safety of distiller grains for animal feed, we conducted a metabolomic analysis of distiller grains during the fermentation process and employed both multidimensional and univariate analyses to screen for inter-group differential metabolites (Figure 4). A total of 4737 metabolites were detected in the pre- and post-fermentation treatments. SJZ9, SJZ30, and SJZ60 exhibited 773 (343 upregulated), 740 (353 upregulated), and 780 (349 upregulated) differential metabolites, respectively, as compared to SJZ0. In terms of the number of differential metabolites, there were relatively few changes in the metabolites during the fermentation of distiller grains; however, the levels of the upregulated compounds gradually shifted toward early fermentation levels as fermentation progressed. This change correlates with our earlier observation that the microbial community structure in the late fermentation stage shifts toward pre-fermentation levels.

Based on the Kyoto Encyclopedia of Genes and Genomes (KEGG) significant enrichment analysis, Tyrosine metabolism and the biosynthesis of unsaturated fatty acids are widely active metabolic pathways in fermented distiller grains (*p* < 0.01, Figure 5). The metabolites related to these pathways are listed in Table 2. Tyrosine serves multiple metabolic pathways in organisms. It is utilized for protein synthesis and is also a precursor for substances such as Thyroxine and Dopamine. Among the biomarkers identified for Tyrosine metabolism, 4-Hydroxyphenylpyruvic acid is a crucial intermediate [23,24]. Tyramine and Dopamine are biogenic amines commonly found in fermented foods; while moderate intake is beneficial, excessive concentrations can impact food flavor and deplete monoamine oxidase, disrupting normal biogenic amine metabolism and leading to adverse effects [25]. Reduced levels of Tyramine and Dopamine metabolism are associated with the *Lactobacillus* genus, which is known for its ability to degrade various biogenic amines, such as Histamine, Tyramine, and Putrescine, across different fermentation systems [26].

The biosynthesis of unsaturated fatty acids is an extensive metabolic activity of the bacterial community during the 9th and 30th days of distiller grain fermentation. Polyunsaturated fatty acids are classified into two main groups according to their structure: n-3PUFAs, which mainly include a-Linolenic acid, Eicosapentaenoic acid, and Clupanodonic acid; and n-6PUFAs, which mainly include Linoleic acid, γ-Linolenic acid, and Arachidonic acid [27]. Long-chain unsaturated fatty acids have important physiological functions. For instance, Docosapentaenoic and Docosahexaenoic acids are the primary active factors in the nutritional function of food, positively impacting the prevention of cardiovascular diseases, alleviating inflammation, and enhancing immune function [28,29]. However, γ-Linolenic acid and Arachidonic acid were not significantly upregulated after 60 days of fermentation. Clearly, an excessively long fermentation time is not a necessary condition for the high-quality fermentation of distiller grains.

Moreover, polyunsaturated fatty acids, which are essential for animals, must be obtained from food sources. Yeasts can also synthesize fatty acids [30,31] (. Distiller grains contain abundant yeast nutrients, such as proteins and oligosaccharides, which support yeast growth. Yeasts and lactobacilli synergistically assimilate lactate and hydrolyze glucosides, serving as metabolic substrates for hetero-fermentative lactobacilli to produce acetic acid [32]. Although animal experiments validating the effects of consuming polyunsaturated fatty acids on organisms are lacking, human studies have suggested that these fatty acids have beneficial effects on animals.

In conclusion, fermentation promotes the degradation of biogenic amines and the formation of polyunsaturated fatty acids in distiller grains, but this promoting effect is limited to short- to medium-term fermentation, as beneficial metabolites (polyunsaturated fatty acids) gradually revert to their unfermented state in the later stages of fermentation.

## 3. Materials and Methods

### 3.1. Fermentation Material and Sample Collection

The distiller grains used in this study were sourced from the Guizhou Maotai Group in Maotai Town, Renhuai City, Guizhou Province, China. The distiller grains mainly consisted of distilled sorghum and wheat, which are byproducts of the brewing process. The microbial fermentation agent was supplied by Yijiayi Bioengineering Co., Ltd., Shijiazhuang, China, and included lactic acid bacteria, yeast, bacillus, *Bifidobacterium*, butyric acid bacteria, amylase, protease, cellulase, and lipase. Initially, the microbial fermentation agent was added to molasses water for ambient temperature fermentation. The specific steps were: A total 100 g of microbial fermentation agent was added to 0.5 kg of molasses mixed with 5 L of water to prepare the molasses water. Subsequently, after 3–5 days of fermentation, the resulting mixture was used to ferment 1 t of distiller grains. The distiller grains were mixed with cornmeal, rapeseed meal, and bran in specified proportions (distiller grains/cornmeal/rapeseed meal/bran = 92%: 3%: 3%: 2%). The prepared molasses water and microbial fermentation agent were added using a shovel, followed by thorough mixing. Samples were collected for analysis after fermentation at room temperature (15–25 °C) for 0, 9, 30, and 60 days. Each bag contained five composite samples collected from different areas, resulting in 24 bags of samples (six replicates × four stages).

### 3.2. Microbiological Analysis

Genomic DNA of the samples was extracted using a DNeasy PowerSoil kit (QIAGEN, Hilden, Germany). The concentration of DNA was detected by 1% agarose gel electrophoresis and NanoDrop2000. The extracted genomic DNA was used as a template for the identification of bacterial diversity based on the selection of sequencing regions using specific primers corresponding to the regions: 16S V3-V4 region (primers 343F: 5′-TACGGRAGGCAGCAG-3′; 798R: 5′-AGGGTATCTAATCCT-3′), and Bio-rad’s Tks Gflex DNA Polymerase (580BR10905) was used for PCR to ensure amplification efficiency and accuracy. PCR system consisted of 2xGflex PCR Buffer (15 μL), 5 pmol/μL primer F (1 μL), 5 pmol/μL primer R (1 μL), template DNA (1 μL), 1.25 U/μL Tks Gflex DNA Polymerase (0.6 μL), and H2O (11.4 μL), which amounted to a total of 30 μL. PCR reaction conditions were as follows: 94 °C pre-denaturation for 5 min; 94 °C denaturation for 30 s, 56 °C annealing for 30 s, and 72 °C extension for 20 s, total 26 cycles; and 72 °C extension for 5 min. After PCR, the products were sent to Ouyi Biomedical Ltd. (Shanghai, China) for sequencing using a MiSeq PE300 platform (Illumina, San Diego, CA, USA).

### 3.3. Metabolomic Analysis

The samples were analyzed for macro-metabolomics. The LC–MS analytical instrument was a liquid–liquid mass spectrometry system consisting of an ACQUITY UPLCI-Class ultra-high-performance liquid-phase tandem VION IMS Q-Tof high-resolution mass spectrometer from Waters (St, Rydalmere, NSW, Australian). The chromatographic conditions were as follows: column ACQUITY UPLC BEH C18 (100 mm × 2.1 mm, 1.7 μm); column temperature, 45 °C; mobile phase A was 0.1% formic acid in water and B was 0.1% formic acid in acetonitrile; flow rate, 0.4 mL/min; injection volume, 1 μL. The sample’s mass spectra were acquired in positive and negative ion scanning mode, and the ion source was electron spray ionization (ESI). The mass spectrometry parameters were 2.5 kV for the electrospray capillary, 40 V for the injection voltage, 4 eV for the collision voltage, 115 °C for the ion source, 450 m for the desolvation temperature, 900 L/h for the carrier gas flow rate, 50–1000 amu for the mass spectrometry scan range, 0.2 s for the scan time, and 0.02 s for the interval.

### 3.4. Bacterial Phenotype Prediction Analyses

The BugBase algorithm (https://bugbase.cs.umn.edu/index.html) was used to predict the phenotypes of bacterial communities, accessed on 28 May 2024. BugBase first normalized the operational taxonomic unit (OTU) using the predicted 16S rRNA copy number and then predicted the microbial phenotype using the pre-calculated files provided. The BugBase algorithm relies on databases, such as Integrated Microbial Genomes (IMG), Kyoto Encyclopedia of Genes and Genomes (KEGG), and Pathosystems Resource Integration Center (PATRIC), to predict phenotypes and corresponding microbial contributors at a genus level.

### 3.5. Statistical Analysis

The study used Microsoft Excel and GraphPad Prism 9.0 software. Using GraphPad Prism 9.0, we conducted statistical analyses to determine the significance of bacterial abundance and phenotypes, with a threshold *p* value of less than 0.05 considered statistically significant.

## 4. Conclusions

Fermentation reduces the pathogenic potential of distiller grains, promotes the growth of probiotics, degrades excessive biogenic amines, increases the content of unsaturated fatty acids, significantly improves the palatability of distiller grains, and extends their shelf life. However, prolonged storage time increases the pathogenic risk and reduces the content of beneficial metabolites. In production practices, the adverse effects of storage time should be considered to reduce the pathogenic risk during animal production. We recommend that distiller grains be fed to animals within 30 d of fermentation. A detailed understanding of the microbial dynamics can be applied directly in practice. Simple tests, such as fermentation time tracking or microbial community analysis, allow producers to quickly assess product safety, providing a cost-effective and efficient method to manage fermentation processes and prevent pathogen proliferation. However, it is important to note that this study was conducted with a single fermentation batch. Therefore, caution should be exercised when generalizing these findings, and future research with biological replicates is needed to confirm the reproducibility of these results.

## Figures and Tables

**Figure 1 ijms-25-11463-f001:**
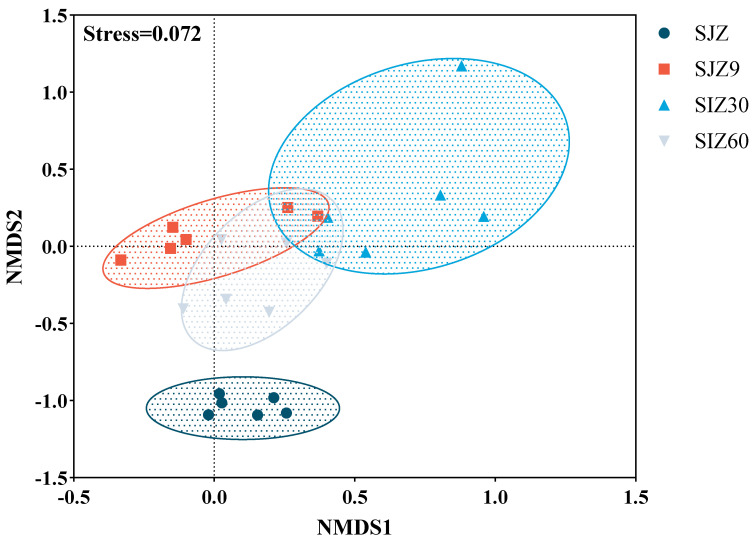
Non-metric multidimensional scaling (NMDS) analysis of bacterial communities. Good model fit was indicated when Stress < 0.2; SJZ: unfermented distiller grain; SJZ9: fermented for 9 d; SJZ30: fermented for 30 d; SJZ60: fermented for 60 d.

**Figure 2 ijms-25-11463-f002:**
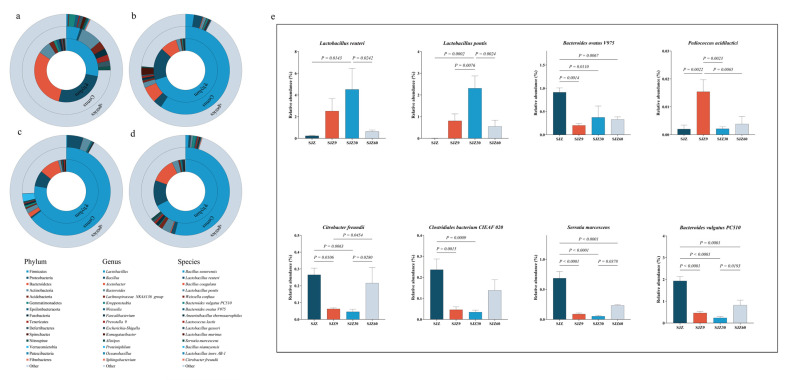
Bacterial community abundance at different fermentation stages of distiller grain. (**a**) Circle plot of bacterial community abundance from gate (internal) to species (external) level in the SJZ; (**b**) Circle plot of bacterial community abundance from gate (internal) to species (external) level in the SJZ9; (**c**) Circle plot of bacterial community abundance from gate (internal) to species (external) level in the SJZ30; (**d**) Circle plot of bacterial community abundance from gate (internal) to species (external) level in the SJZ60; (**e**) Relative abundance of representative species. SJZ: unfermented distiller grains; SJZ9: fermented for 9 d; SJZ30: fermented for 30 d; SJZ60: fermented for 60 d.

**Figure 3 ijms-25-11463-f003:**
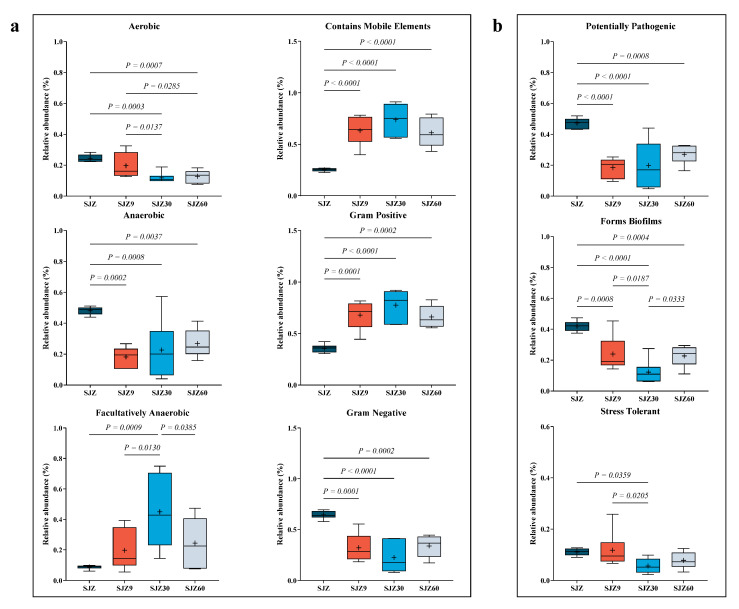
Annotation of bacterial phenotypes at different fermentation stages of wine distiller grain. (**a**) Bacterial characterization of distiller grains at different fermentation times; (**b**) Phenotypes reflecting bacterial resistance at different fermentation stages of wine distiller grains. SJZ: unfermented distiller grain; SJZ9: fermented for 9 d; SJZ30: fermented for 30 d; SJZ60: fermented for 60 d.

**Figure 4 ijms-25-11463-f004:**
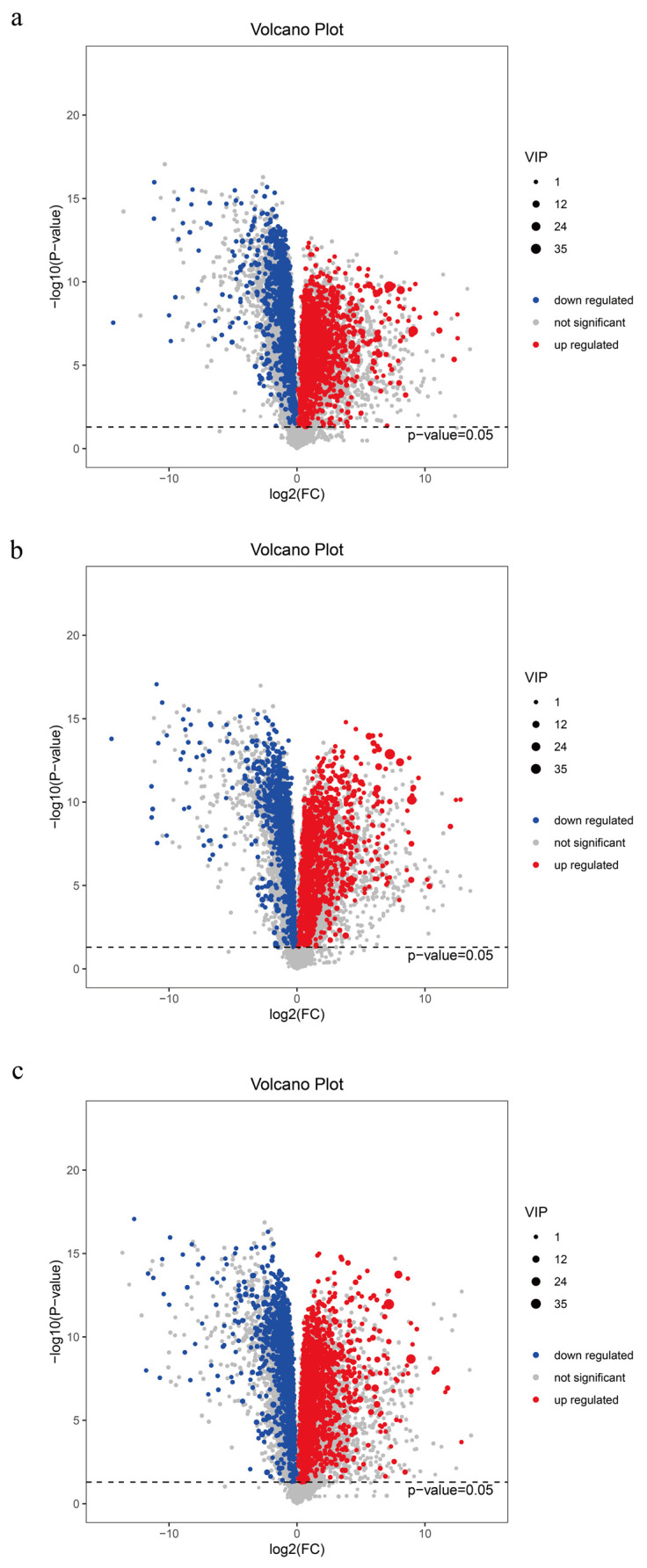
Differential metabolites of distiller grains at different stages of fermentation. (**a**) SJZ9 vs. SJZ; (**b**) SJZ30 vs. SJZ; (**c**) SJZ60. vs. SJZ. In the volcano plots, the red origin represents significantly upregulated metabolites in the experimental group, the blue origin represents significantly downregulated metabolites, and the gray point represents insignificant metabolites. The dotted line indicates the significance level threshold (*p* = 0.05).

**Figure 5 ijms-25-11463-f005:**
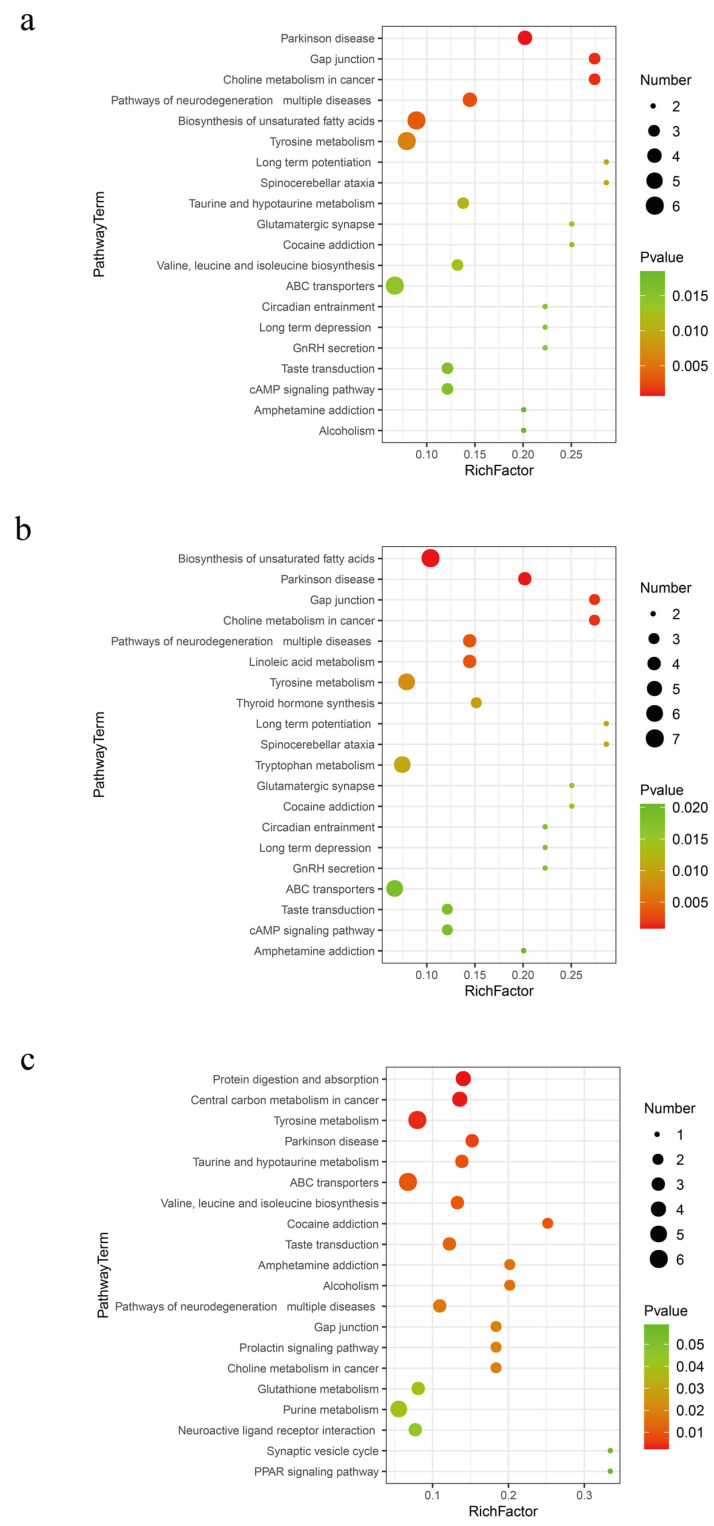
Metabolic pathways in different fermentation stages of distiller grains. (**a**) SJZ9 vs. SJZ; (**b**) SJZ30 vs. SJZ; (**c**) SJZ60 vs. SJZ.

**Table 1 ijms-25-11463-t001:** Alpha diversity during distiller grain fermentation.

Item	Community Richness		Community Diversity		Goods Coverage
	Chao1	Observed Species	Shannon	Simpson	
SJZ	3075.70	1572.40	8.59	0.99	0.99
SJZ9	2167.61	1384.45	4.32	0.81	0.99
SJZ30	1823.92	1247.37	4.77	0.88	0.99
SJZ60	2320.42	1354.13	5.32	0.82	0.99

**Table 2 ijms-25-11463-t002:** Differential metabolites associated with Tyrosine metabolism and biosynthesis of unsaturated fatty acids.

Metabolites	SJZ9 vs. SJZ				SJZ30 VSS JZ				SJZ60 vs. SJZ			
	VIP	log2(FC)	*p*-Value	Trend	VIP	log2(FC)	*p*-Value	Trend	VIP	log2(FC)	*p*-Value	Trend
Tyramine	1.05	−1.61	<0.001	Downregulated	1.02	−1.4	<0.001	Downregulated	1.09	−1.66	<0.001	Downregulated
Dopamine	1.06	−1.68	<0.001	Downregulated	1.08	−1.7	<0.001	Downregulated	1.07	−1.58	<0.001	Downregulated
4-Hydroxyphenylpyruvic acid	1.17	1.47	<0.001	Upregulated	0.99	1.19	<0.001	Upregulated	1.15	1.4	<0.001	Upregulated
Clupanodonic acid	1.41	7.91	<0.001	Upregulated	1.44	7.99	<0.001	Upregulated	1.14	7.34	<0.001	Upregulated
Arachidic acid	1.01	0.71	0.027	Upregulated	1.51	0.99	<0.001	Upregulated	0.36	0.25	0.452	-
Gamma-Linolenic acid	2.24	0.97	0.031	Upregulated	2.02	0.64	<0.001	Upregulated	0.57	0.16	0.347	-

## Data Availability

The data presented in this study are available on request from the corresponding author.

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
