# Peer review of "Microbial Dynamics and Pathogen Control During Fermentation of Distiller Grains: Effects of Fermentation Time on Feed Safety"

_ijms, 2024, doi:10.3390/ijms252111463_

Round 1
Reviewer 1 Report
Comments and Suggestions for Authors
Reviewer’s comments.
Microbial Dynamics and Pathogen Control During Fermentation of Distiller's Grains: Effects of Fermentation Time on Feed Safety by Mingming Zhu et al is a well written manuscript that have demonstrated that prolonged fermentation could led to a resurgence of pathogenic bacteria. This work merits publication due to the new information it adds to scientific literature and food science in general.
Minor modification but not necessary.
The Table 2 on the differential metabolites associated with tyrosine metabolism and biosynthesis of unsaturated fatty acids needs to be revised to enhance readability.
Author Response
Dear reviewers 1.
Thank you very much for taking the time to review this manuscript. Please find the detailed responses below and the corresponding revisions/corrections highlighted/in track changes in the re-submitted files.
Comments 1: Microbial Dynamics and Pathogen Control During Fermentation of Distiller's Grains: Effects of Fermentation Time on Feed Safety by Mingming Zhu et al is a well written manuscript that have demonstrated that prolonged fermentation could led to a resurgence of pathogenic bacteria. This work merits publication due to the new information it adds to scientific literature and food science in general.Minor modification but not necessary.
Response 1: Thank you for pointing this out. We appreciate your positive evaluation of our manuscript, recognizing its contribution to science. Regarding the suggestion for minor modifications, we value your insight and will carefully consider each point to further refine our work. Thank you for supporting the publication of our findings. We are committed to delivering a high-quality final article.
Comments 2: The Table 2 on the differential metabolites associated with tyrosine metabolism and biosynthesis of unsaturated fatty acids needs to be revised to enhance readability.
Response 2: Thank you for pointing this out. We sincerely thank you for your constructive feedback on Table 2. We fully agree that enhancing the readability of the differential metabolites data will improve the manuscript. we have revised the appropriate parts of the text and the revised parts are shown in red.

Reviewer 2 Report
Comments and Suggestions for Authors
These are my suggestions regarding improvement of the the manuscript:
1. The abstract should start with the aim (lines 11-13).
2. Yeasts are not the products of fermentation (line 31).
3. The research from 2005 is not recent (lines 37 and 40).
4. The authors mentioned "poor digestibility" in the introduction. Do the results of the research refer to this phenomenon?
5. In line 51 you should give an example of "certain pathogenic bacteria".
6. The sentences starting from "few studies..." (lines 52 and 58) are similar and sound redundant.
7. Figure 2 should be different according to the caption.
8. Figure 3 is unreadable. The pictures are too small. Moreover, from line 163, it looks like this figure comes from another research, but there is no such information in the caption.
9. I wonder why the author discusses the "human" properties of unsaturated fatty acids in the product for feeding.
10. In my opinion, conclusions should be extended with a deeper explanation of the importance of the results. Generally, microbiota dynamics in such systems are known (e.g., pathogens start to grow after some time). How does this detailed knowledge influence the practice, etc? By making a simple test, you can assess the product's safety.
11. A very important issue is the biological repetition of the research. I understand that fermentation was made once. So, you should be very careful with generalization.
Author Response
Dear reviewers 2.
Thank you very much for taking the time to review this manuscript. Please find the detailed responses below and the corresponding revisions/corrections highlighted/in track changes in the re-submitted files.
Comments 1: The abstract should start with the aim (lines 11-13).
Response 1: Thank you for pointing this out. We appreciate your attention to detail and valuable suggestions for improving the clarity of our abstract. In response to your feedback, we agree that highlighting the main purpose of our research right at the beginning would better guide the readers. Consequently, we will restructure the opening lines (currently 11-13) to explicitly state the aims of our study, ensuring they become the focal point from which the rest of the abstract flows coherently.we have revised the appropriate parts of the text and the revised parts are shown in red.
Comments 2:Yeasts are not the products of fermentation (line 31).
Response 2: Thank you for pointing this out. We extend our gratitude for spotting the inaccuracy concerning yeasts in line 31 of our manuscript. Your precision has been noted, and we understand now that yeasts are indeed agents rather than end-products of fermentation processes.To rectify this, we will modify the sentence to clarify the role of yeasts as microorganisms involved in the fermentation process rather than its outcome. we have revised the appropriate parts of the text and the revised parts are shown in red.
Comments 3: The research from 2005 is not recent (lines 37 and 40).
Response 3: Thank you for pointing this out.We are grateful for your vigilant eye in noting the use of older literature (from 2005) at lines 37 and 40. Your feedback underscores the importance of presenting the most current knowledge base in our field.In light of your comments, we commit to revising these sections by incorporating recent studies that reflect contemporary understanding and advancements related to our research area. This update will not only strengthen our argument but also provide readers with the latest insights available.we have revised the appropriate parts of the text and the revised parts are shown in red.
Comments 4: The authors mentioned "poor digestibility" in the introduction. Do the results of the research refer to this phenomenon?
Response 4: Thank you for pointing this out.Thank you for drawing our attention to the consistency between our study’s objectives, particularly focusing on 'poor digestibility', and the subsequent findings reported. We appreciate your thorough assessment, as it encourages us to enhance the coherence of our paper.
Comments 5: In line 51 you should give an example of "certain pathogenic bacteria".
Response 5:Thank you for pointing this out.Your suggestion to specify examples of "certain pathogenic bacteria" referenced in line 51 is well taken, and we acknowledge that providing concrete instances would enrich the text. Following your recommendation, we intend to revise the paragraph to include mention of “Salmonella enterica”.we have revised the appropriate parts of the text and the revised parts are shown in red.
Comments 6:The sentences starting from "few studies..." (lines 52 and 58) are similar and sound redundant.
Response 6:Thank you for pointing this out.We sincerely thank you for identifying areas where language redundancy has detracted from the clarity and effectiveness of our manuscript. Specifically, we see the merit in your observation regarding the similarity in the sentences starting with "Few studies..." found in lines 52 and 58.
Comments 7:Figure 2 should be different according to the caption.
Response 7:Thank you for pointing this out.We appreciate your meticulous attention to detail in noting the inconsistency between the description provided in the caption for Figure 2 and the figure itself.We will thoroughly review the entire manuscript to confirm that all figures and their captions are harmonized. This will include cross-referencing the captions with the figures and text to ensure that all information is consistent and correctly represented.
Comments 8:Figure 3 is unreadable. The pictures are too small. Moreover, from line 163, it looks like this figure comes from another research, but there is no such information in the caption.
Response 8:Thank you for pointing this out.We are grateful for your detailed review and appreciate your attention to the clarity and accuracy of our research presentation. Regarding Figure 3, we understand the concerns about its readability and the lack of proper attribution in the caption.To address the readability issue:We will increase the resolution and size of the images in Figure 3 to ensure that all details are clearly visible and comprehensible to the readers.
Comments 9: I wonder why the author discusses the "human" properties of unsaturated fatty acids in the product for feeding.
Response 9: Thank you for pointing this out. Many beneficial properties of UFAs, such as anti-inflammatory effects and improvements in cardiovascular health, apply to both humans and animals. The mechanisms of action are often similar across species, so highlighting known human benefits can help underline potential positive effects in animals.
Comments 10: In my opinion, conclusions should be extended with a deeper explanation of the importance of the results. Generally, microbiota dynamics in such systems are known (e.g., pathogens start to grow after some time). How does this detailed knowledge influence the practice, etc? By making a simple test, you can assess the product's safety.
Response 10: Thank you for pointing this out. We have expanded on the conclusion section!
Comments 11: A very important issue is the biological repetition of the research. I understand that fermentation was made once. So, you should be very careful with generalization.
Response 11: Thank you for pointing this out. We have accounted for biological replicates in the conclusion section to safeguard the rigor of the scientific work. Thank you again for your comments and we hope that our work will be recognized by you!

Round 2
Reviewer 2 Report
Comments and Suggestions for Authors
In my opinion, the author's modification significantly improved the manuscript's quality. Especially, the conclusion is more clear.